# A developmental atlas of zebrafish gills links early vascular patterning to adult architecture

Mathieu Preußner[1], Anna Mertens[1], Marion Basoglu[2] and Virginie Lecaudey[1,*]

## ABSTRACT

Gills are essential for fish respiration and have a highly specialized cellular architecture enabling efficient gas exchange. Surprisingly, the developmental processes underlying gill formation in zebrafish remain poorly understood. Here, we present for the first time a comprehensive analysis of the morphogenesis of gill arteries, filaments and lamellae during lifelong development. Our results provide important insights into the temporal and spatial pattern of gill angiogenesis, revealing fundamental differences in the formation of lateral versus medial filaments along the dorso-ventral axis. These early asymmetries correlate with, and likely underlie, the structural asymmetries observed in adult gills, which we quantitatively characterize. This indicates that a region-specific developmental programme establishes a blueprint for gill architecture maintained throughout life. We further show that lamellae develop through a complex interplay between endothelial and cranial neural crest-derived pillar cells. Notably, lamellar size, which strongly influences respiratory efficiency, depends on the position of the filament in the arch. Together, our work identifies key cellular and temporal mechanisms driving gill development, and provides a framework to investigate broader principles of branching morphogenesis and angiogenesis in vertebrates.

KEY WORDS: Gills, Zebrafish, Pillar cells, Branching, Angiogenesis, Morphology

## INTRODUCTION

Branching morphogenesis is a fundamental process in animal development, underlying the formation of complex organs such as the lung, kidney and vascular networks (Goodwin and Nelson, 2020). Gills represent another highly branched organ, serving as the primary respiratory surface in fish, by facilitating oxygen uptake from the surrounding water. Like the lung, gills optimize gas exchange by maximizing surface area through extensive epithelial and vascular branching, while maintaining direct exposure to the environment.

In teleosts, the largest group of ray-finned fish, including *Danio rerio*, the gills consist of four gill arches on each side, covered by a

protective bony plate known as the operculum (Fig. 1A). Each gill arch is supplied by a branchial artery (BA) originating from the ventral aorta and delivering blood under pressure. From each gill arch, two alternating rows of filaments extend laterally, each row constituting a hemibranch and a set forming a holobranch (Fig. 1A,E) (Evans et al., 2005). Two major vessels run along each filament, the afferent filament artery (AFA) carrying deoxygenated blood, and the efferent filament artery (EFA) carrying oxygenated blood. These vessels branch repeatedly along the length of the filament to supply each lamella (Fig. 1E).

Each filament supports numerous lamellae, which serve as the primary sites of gas exchange. Lamellae consist of two highly specialized epithelial sheets separated by regularly spaced, endothelial-like pillar cells (PCs) (Morgan, 1974; Newstead, 1967). The spaces between adjacent PCs form a shared vascular compartment within the lamella (Evans et al., 2005; Kato et al., 2007; Wilson and Laurent, 2002) (Fig. 1H). This vascular space is delimited on the proximal side by the filament and distally by the endothelial outer marginal channel (OMC), which spans from the AFA to the EFA. As erythrocytes traverse this vascular space, they capture oxygen from the water circulating between the lamellae, a process facilitated by the covering epithelium (Kato et al., 2007; Morgan and Tovell, 1973; Wilson and Laurent, 2002).

Early studies from the 20th century characterized the structural arrangement and provided insights into gill development in various fish species (Bettex-Galland and Hughes, 1973; Kimmel et al., 1995; Morgan, 1974; Wilson and Laurent, 2002). More recent work employing cell lineage tracing and single-cell RNA-sequencing has advanced our understanding of the complex molecular and cellular processes underlying gill development, regeneration and immune function (Dalum et al., 2021; Fabian et al., 2022; Mongera et al., 2013; Stolper et al., 2019). These studies have shown that gills develop on parts of the pharyngeal apparatus during early embryogenesis (Gillis and Tidswell, 2017; Mongera et al., 2013). Subsequently, the respiratory organ emerges through coordinated contributions from mesodermal, endodermal and ectodermal cells (Gillis and Tidswell, 2017; Warga and Nüsslein-Volhard, 1999). At the distal growing tip of the filament, cranial neural crest-derived cells form a pool of gill progenitors that contribute to the endothelial-like PCs, the gill cartilage and the tunica media around the large filament vessel (Fabian et al., 2022; Mongera et al., 2013; Stolper et al., 2019). Despite these advances, the developmental dynamics and spatial organization of the vascular network and of the major gill cell types remain poorly understood, limiting broader use of this complex system.

To fill this gap, we combined state-of-the-art live microscopy, high-resolution imaging of cleared tissue and electron microscopy to provide the first comprehensive, high-resolution and quantitative framework for gill development. Our analysis uncovered a surprisingly complex morphogenetic process characterized by asymmetries in filament formation along the dorso-ventral and medio-lateral axes of

[1]Department of Developmental Biology of Vertebrates, Institute of Cell Biology and Neuroscience, FB15, Goethe Universität Frankfurt, Max-von-Laue-Str. 13, 60438 Frankfurt am Main, Germany. [2]Electron Microscopy Facility, FB15, Goethe Universität Frankfurt, Max-von-Laue-Str. 13, 60438 Frankfurt am Main, Germany.

*Author for correspondence (lecaudey@bio.uni-frankfurt.de)

M.P., 0000-0003-2931-7717; V.L., 0000-0002-8713-3425

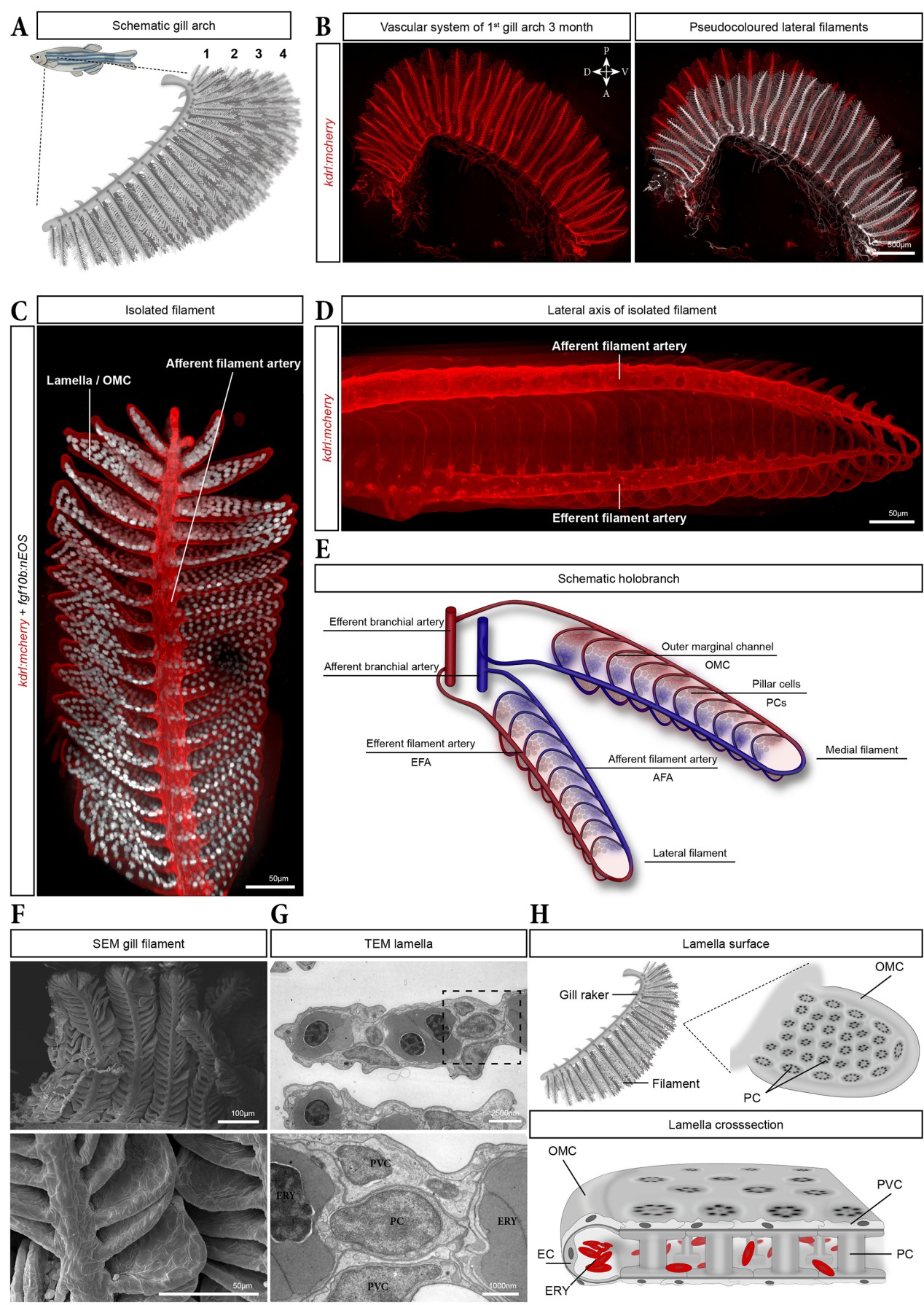

**Fig. 1.** See next page for legend.

**Fig. 1. Gill architecture and vascular anatomy in adult zebrafish.**
(A) Schematic of the four gill arches of an adult zebrafish. (B) Vascular network of a first gill arch isolated from a 3-month-old *kdrl:mCherry* fish and cleared using the CUBIC method, with the lateral hemibranch pseudo-coloured in white (right panel). (C) Digitally isolated gill filament from an adult zebrafish expressing *kdrl:mCherry* in endothelial cells (ECs) (red) and *fgf10b:nEOS* in pillar cells (PCs) (white). (D) Lateral view of an isolated filament in an adult *kdrl:mCherry* zebrafish. (E) Schematic showing the vasculature in a pair of filaments. Red and blue vessels represent oxygen-rich and oxygen-poor blood, respectively. (F) SEM images of gill filaments and lamellae at two different magnifications. (G) TEM images of ultrathin sections of a lamella. The close-up (lower panel) shows a PC nucleus surrounded by two pavement cells (PVCs) above and below and two erythrocytes (ERY) left and right. (H) Schematic of an isolated gill rake (top left) and lamella (top right), illustrating the regularly organized PCs and the outer marginal channel (OMC), and a cross-section (bottom) of a lamella showing erythrocyte flow within the vascular space bordered by PCs and ECs of the OMC. For all figures, unless otherwise indicated, fluorescent images are maximum intensity projections (MIPs) of z-stacks of the gills in *Tg(kdrl:mCherry)* transgenic zebrafish acquired using a spinning-disc confocal microscope. In the overview images showing all four gill arches, every other gill arch has been pseudo-coloured to distinguish individual arches and the arches are numbered 1 to 4; numbers indicate the arch, isolated digitally using Imaris, shown in adjacent panels.

the gill arches. Notably, we show that this precise developmental pattern lays the foundation for the characteristic adult gill architecture. Our study further establishes the zebrafish gill as a valuable model for studying the molecular and cellular mechanisms underlying branching morphogenesis, endothelial tip cell behaviour, vascular anastomosis and PC-endothelial cell (EC) interactions. We anticipate that this resource will serve as a foundation for future functional and mechanistic studies.

## RESULTS
### Gill anatomy and vascular architecture reveal medio-lateral asymmetries in filaments
To characterize the overall morphology and vascular organization of adult zebrafish gills, we performed high-resolution imaging on whole, cleared gills using the *Tg(kdrl:ras-mcherry)^s896* (hereafter *kdrl:mCherry*), which labels EC membranes and thus delineates the entire gill vascular network. This revealed a consistent medio-lateral asymmetry in filament length along the dorso-ventral axis: medial filaments were markedly longer than their lateral counterparts (pseudo-coloured white), except in the ventral-most region of the arch where lateral filaments were slightly longer (Fig. 1B).

To examine the morphology of a filament in more detail, we combined *kdrl:mcherry* with the recently-described *Tg(fgf10b: nEOS)^el865* (hereafter *fgf10b:nEOS*) reporter line (Fabian et al., 2022). The *kdrl:mCherry* signal highlighted AFAs branching from the afferent BA and supplying each filament (Fig. 1B,E). A vascular loop extended distally and drained into the EFA (Fig. 1C,D, schematic in E). Dozens of flat, sheet-like lamellae branched bilaterally from each filament main axis (Fig. 1C). *kdrl:mCherry* was only expressed in the ECs of the OMC, forming a half-tube-shaped channel at the distal end of the lamella, while *fgf10b:nEOS* highlighted the regular organisation of the PC nuclei within the lamella (Fig. 1C).

Finally, to investigate the ultrastructure of the lamellae, we performed electron microscopy on adult gills. Scanning electron microscopy (SEM) confirmed the overall gill architecture, with each arch composed of parallel, densely packed filaments, themselves covered with lamellae (Fig. 1F). The whole filament, including the lamellae, was covered by epithelial pavement cells (PVCs) forming a thin epithelial layer (Fig. 1F) (Laurent and Dunel, 1980).

Transmission electron microscopy (TEM) further revealed the cellular organization within the lamellae (Fig. 1G,H). PCs were regularly spaced within the lamellae and exhibited their characteristic morphology, with a central cell body extending flat, cytoplasmic arms (Fig. 1G,H) that made close contact with those of adjacent PCs, thereby forming the vascular space (Fig. 1G,H), filled with nucleated erythrocytes (ERY). PCs were flanked on both sides by PVCs, with their nuclei generally flanking the PC bodies (Fig. 1G). At the distal end of the lamella, PCs cytoplasmic arms were in direct contact with an EC, which formed the OMC (Fig. 1G,H).

Together, these complementary imaging approaches revealed the detailed cellular and vascular architecture of the adult gill and uncovered a consistent medio-lateral asymmetry in filament length across all arches.

### Onset of gill filament formation and lateral branchial artery
To determine how these spatial asymmetries emerge during gill development, we first performed a time series of the four gill arches using confocal microscopy with fixed *kdrl:mCherry* embryos and larvae between 2 and 10 days post-fertilization (dpf) (Fig. 2). At 2.5 dpf, the branchial vascular system consisted of four thin filament-free BAs, representing the early vascular structures of the prospective gill arches (Fig. 2A). At 3.5 dpf, the BAs had become larger and the first filaments were visible, with a clear maturation gradient from BA1 to BA4 (Fig. 2B). In the following days, the four BAs elongated and additional filaments formed on each (Fig. 2C-E). Already in these early stages, two types of differently-oriented filaments could be observed, indicating differences in their formation (Fig. 2B-E). Medial filaments (arrowheads) were closer to the medial axis of the fish, while lateral filaments (arrows) were closer to the operculum. Anterior to the four BAs, the operculum artery (wide arrowhead in Fig. 2A,C-E) partially covers BA1. Analysing this time series revealed the complexity of gill vascular development, prompting us to dissect the process in detail, focusing on BA1, the most mature BA, as a reference.

Spinning-disc live-cell microscopy of *kdrl:mCherry* embryos transitioning from 2.5 to 3 dpf revealed that BAs expanded their diameter and their length significantly during this period (Fig. S1). In timelapses starting at 3 dpf, we observed a tip cell sprouting from the BA approximatively midway along the dorso-ventral axis (Fig. 3A, arrowhead 1). This tip cell initially migrated posteriorly, perpendicular to the BA, before reorienting to form a small loop, thereby creating the first medial filament (Fig. 3A, arrowhead 1 at 03:45 h). Over the next 10 h, two additional medial filaments began to sprout from the BA, ventral and dorsal from the first filament (Fig. 3A, arrowheads 2 and 3). Over time, the tip cell of the first filament reoriented and migrated ventrally, eventually aligning parallel to the BA (highlighted in white in Fig. 3A, 12:30 h). Finally, this tip cell anastomosed with the vascular loop of the adjacent ventral filament (Fig. 3A, 15:50 h), forming a connecting vessel between the two medial filaments (Movie 1). This vessel further extended ventrally and fused with the BA (wide arrowhead in Fig. 3A, 24:00 h). From this point onward, the *de novo* formed vessel will be referred to as the lateral branchial artery (LBA), while the dorsal segment of the original BA, located above the fusion site, will be designated as the medial branchial artery (MBA) (Fig. 3C).

Interestingly, this sequence of LBA formation could also be inferred from single fixed samples by comparing the developmental stages of the four branchial arches (Fig. 3B). At 3.5 dpf, the fourth arch was the least developed, consisting solely of the BA, similar to the first gill arch at 2.5 dpf and beginning of the timelapse (Fig. 2A; Fig. 3A at 0:00 h and Fig. 3B). In the third arch, the first medial

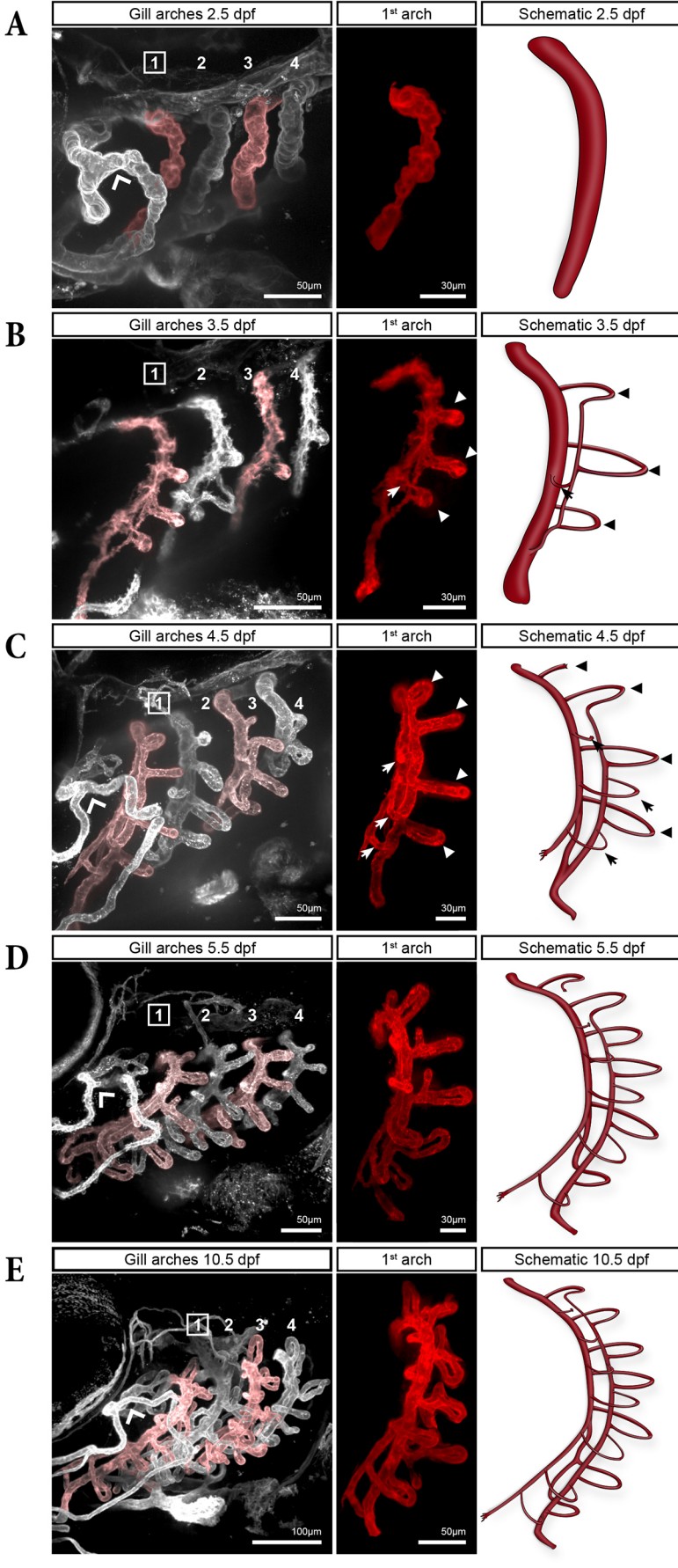

**Fig. 2. Progressive development of gill vasculature in *kdrl: mCherry* zebrafish larvae between 2.5 and 10.5 dpf.** (A-E) For each stage, a maximum intensity projection (MIP) of the four gill arches (left), the corresponding first gill arch digitally isolated (middle) and a schematic of this arch (right) are shown. In the schematics, the lamellae have been omitted for clarity. In B and C, arrowheads and arrows are pointing to medial and lateral filaments, respectively.

filament had formed, while in the second arch, two additional medial filaments were already visible. Finally, in the first arch, the first three medial filaments were already anastomosed to form the LBA, which had already fused with the BA (Fig. 3B).

## Dynamics of medial and lateral filament formation in the dorsal part of the arch

Next, we wanted to further understand the dynamic process of filament formation, in particular the coordination between medial and lateral filament formation after the emergence of the LBA. We therefore performed additional timelapses on 3.5 dpf *kdrl:mCherry*

embryos (Fig. 4). In the dorsal region of the arch, additional filaments consistently formed medially first (Fig. 4A), following a growth pattern similar to the initial outgrowth of medial filaments (Fig. 3). Timelapse microscopy revealed that ECs of the prospective medial filament sprouted from the MBA (Fig. 4A, 00:40 h) and formed a small vascular loop (Fig. 4A, 04:20 h). Over time, the tip cell reoriented and extended ventrally, aligning parallel to the MBA (Fig. 4A, 6:00 h). As the tip cell continued to elongate, it anastomosed with the already formed LBA (Fig. 4A, 08:00 h), resulting in the extension of the LBA dorsally (Movie 2). Notably, this process repeated sequentially for each newly formed medial

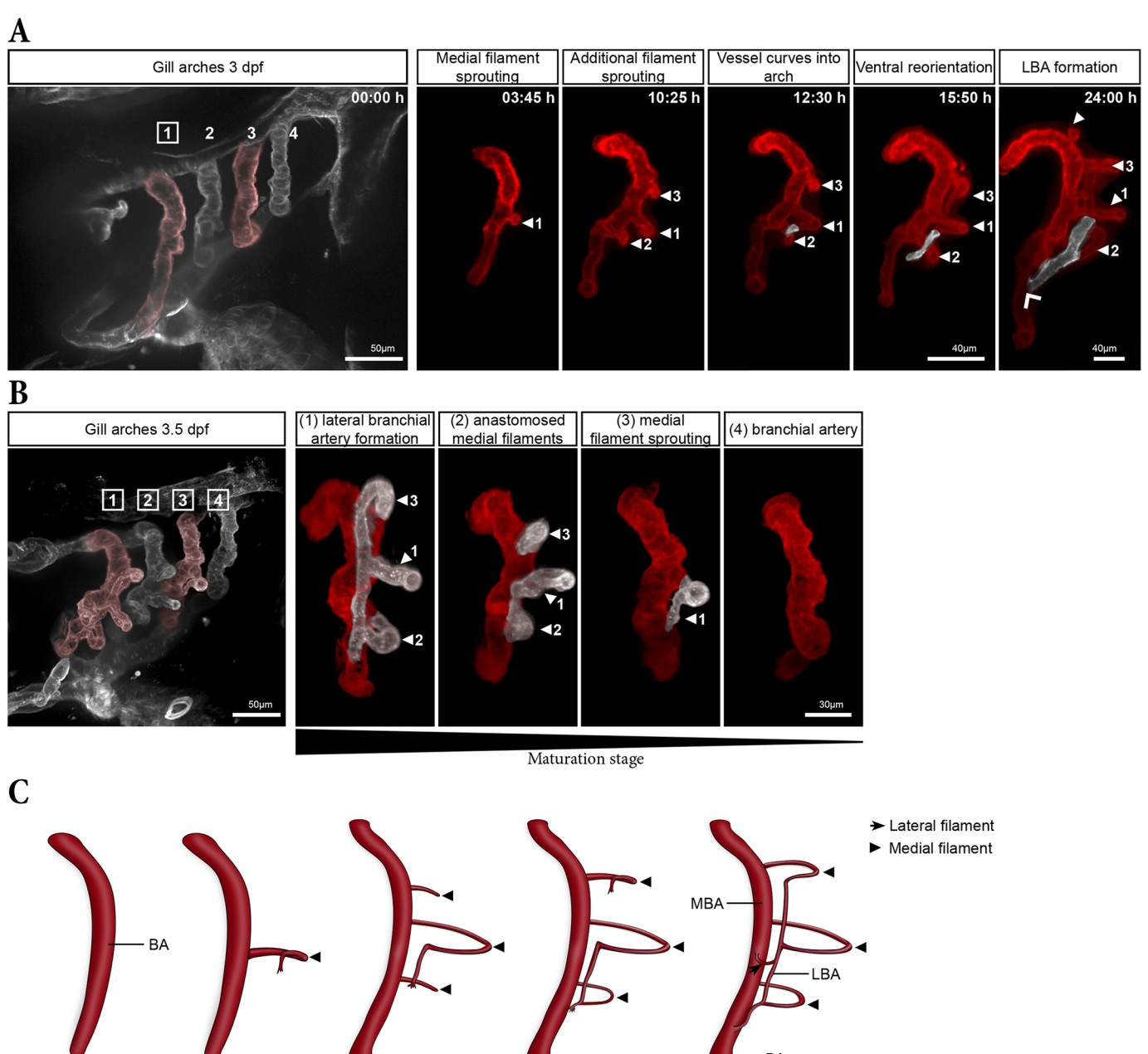

**Fig. 3. Onset of gill filament formation.** (A) Maximum intensity projection (MIP) from a 16-h timelapse of gill vasculature development in a 3 dpf embryo, shown at five time points. The final panel (24:00 h) was acquired 8 h after the timelapse ended. The left-most image shows all four gill arches, while the subsequent panels show only the digitally isolated first arch to highlight the emergence of the first three filaments (numbered arrowheads) and of the lateral branchial artery (LBA) (pseudo-coloured in white). The wide arrowhead indicates the fusion between the LBA and the medial branchial artery (MBA). (B) MIP of the gill arches in a fixed 3.5 dpf embryo, with digitally isolated first to fourth gill arches highlighting the maturation gradient. Medial filaments and the LBA are pseudo-coloured in white. (C) Schematic of the onset of gill filament and LBA formation.

**A**

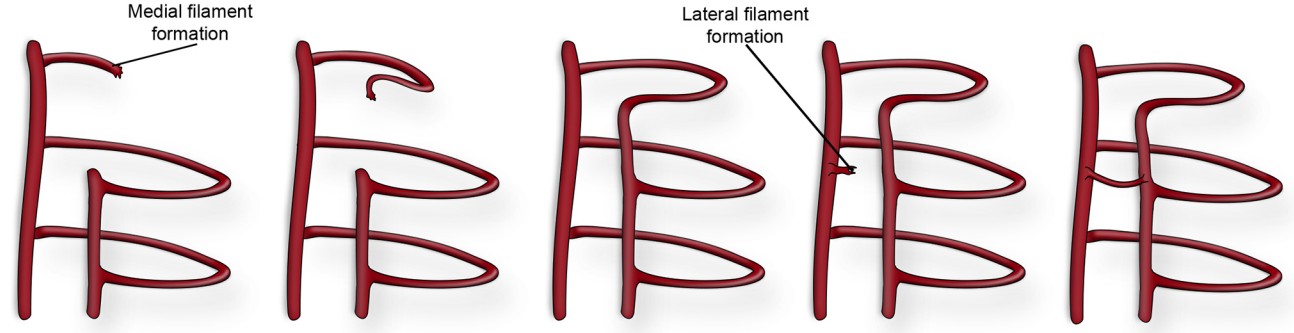

**Fig. 4. Medial and lateral filament formation in the dorsal region of the gill arches.** (A,B) Maximum intensity projection (MIP) from a 15-h timelapse of the gill vasculature in a 3.5 dpf embryo, in which the second gill arch was digitally isolated at given time points to highlight the formation of medial (A) and lateral (B) filaments (pseudo-coloured in white). Dashed box in B highlights the area where the lateral filament starts to sprout (enlarged in the three middle panels). (C) Schematic representation of medial and lateral filament formation in the dorsal region of the gill arch.

filament (arrowhead in Fig. 4A, 14:40 h; Fig. S3A), suggesting a stepwise extension of the LBA dorsally.

Once the extension of the LBA was completed between two medial filaments, a new lateral filament sprouted from the MBA (Fig. 4B, 02:20 h). ECs of the nascent lateral filament formed a vascular bow toward the newly extended LBA segment and fused with it between the two medial filaments (Fig. 4B; Movie 3). The lateral filaments did not form a vascular loop between the MBA and the LBA and were therefore initially much shorter than the medial filaments.

Taken together, these results demonstrate that filament formation in the dorsal region begins with medial filaments, which contribute to the sequential extension of the LBA dorsally (Fig. 4C). Once the LBA was established between two neighbouring medial filaments, a lateral filament emerged from the MBA, connecting to the LBA (Fig. 4C).

### MBA branching and pinch-off break the symmetry of development in the gill arch
Initially we assumed that filament formation and LBA growth proceeded ventrally in the same manner as described previously.

Surprisingly, timelapses of *kdrl:mCherry* embryos transitioning from 4.5 to 5 dpf revealed that filaments formed in a different manner ventral to the LBA-BA fusion point as compared to dorsal (Fig. 5A; Movie 4). Shortly after the LBA had fused with the BA, the MBA started to branch at a slightly more dorsal position relative to the LBA-BA fusion site (asterisk, Fig. 5C). This branch, which then outgrew ventrally, will be referred to as the outer medial branchial artery (OMBA) (Fig. 5C-C″). This branching represents a key symmetry-breaking event, distinguishing dorsal from ventral filament formation. Ventral to this branching point, all lateral filaments successively sprouted from the OMBA and connected with the LBA. The first ventral lateral filament formed a tiny vascular bow extending from the OMBA to the LBA (encircled arrow in Fig. 5C′,C″,D′,D″,E′,E″), similar to the initial dorsal lateral filament (pseudo-coloured in white in Fig. 4B). Interestingly, the medial filament located just dorsal to the branching point (arrowhead in Fig. 5B-E″) served as a reliable marker. Indeed, the immediately adjacent dorsal medial filament consistently corresponded to the very first-formed filament (arrowhead 1 in Fig. 3A) and could still be identified in mature fish (pseudo-coloured filament in Fig. S2). Initially, the other half of the MBA branch, which we refer to as the inner medial branchial artery (IMBA), remained in continuity with the original BA (schematic in Fig. 5B″). Concomitantly with the branching of the MBA, the connection between the IMBA and the BA was pinched (wide arrowhead in Fig. 5B-B″) and then disconnected (wide arrowhead in Fig. 5C′,C″). This event will cause deoxygenated blood coming from the heart to flow only into the LBA and reach the MBA exclusively via the filaments, ensuring its oxygenation (schematic in Fig. 5A). After the pinch-off, the dorsal part of the severed IMBA re-entered a proangiogenic state, and a protruding tip cell became visible and progressed ventrally (wide arrowhead in Fig. 5D, 06:20 h, D′,D″; Movie 4). In the next hours, this new sprout reoriented posteriorly to form a vascular loop and fused with the LBA, creating the very first ventral medial filament (encircled arrowhead in Fig. 5E-E″). In the meantime, filaments continued to form dorsally.

We next examined the gills in fixed 5.5 and 10.5 dpf larvae to further track the ongoing development and arrangement of ventrally developing filaments (Fig. 6). The 5.5 dpf larvae closely aligned with the end of the timelapse in Fig. 5, but showed a more mature IMBA-derived medial filament (encircled arrowhead in Fig. 6A,A′). Additionally, a new lateral filament grew ventrally from the OMBA, marking the most ventral filament (boxed arrow in Fig. 6A′). By 10.5 dpf (Fig. 6B,B′), an additional pair of filaments had formed ventrally. Between 5.5 and 10.5 dpf, a new branch sprouted from the IMBA (wide arrowhead in Fig. 6A′,B′), extended ventrally before reorienting posteriorly to give rise to the subsequent medial filament (boxed arrowhead in Fig. 6B′). In contrast, the newly formed lateral filament (hexagonal arrow in Fig. 6B′) appeared again as a short, straight vessel directly connecting the OMBA to the LBA, very similar to the first ventral lateral filament (encircled arrow in Fig. 6B′). Ventral to the branching point, there was generally one more lateral filament than medial filaments, indicating a developmental delay in medial filaments.

Taken together, our observations demonstrate that the branching of the MBA is a symmetry-breaking event that distinguishes dorsal from ventral filament formation. In the ventral part of the arch, lateral filaments consistently emerge first from the newly established OMBA. In contrast, medial filaments arise exclusively from the IMBA after its separation from the BA and exhibit a developmental delay. Ultimately, these results show two different, stereotyped patterns in the dorsal and the ventral regions of the arch, with all filaments converging onto the LBA.

## Interplay of OMC tip cells and endothelial-like PCs controls lamellae morphology and outgrowth

ECs are known to form the OMC of gill lamellae (Morgan, 1974). However, the precise onset of lamellae development and outgrowth has not yet been described. First signs of developing lamellae, individual endothelial tip cells of the OMC, were observed at 4 dpf in *kdrl:mCherry* embryos, primarily in the first gill arch and occasionally in the second (Fig. S3A′).

To better characterize these events, we focused on fixed 5 dpf larvae (Fig. 7A-B′), where numerous endothelial tip cells were identified in most gill arches, with some already having established OMCs (dashed box in Fig. 7B′). Those sprouting tip cells, originating from the EFA, migrated towards the low oxygenated AFA (Fig. 7A-C). They ultimately formed small lamellae (upper panel in Fig. 7B′,C). Both lateral and medial filaments showed tip cells contributing to the formation of the lamellae by establishing the OMC (pseudo-coloured tip cells in Fig. 7A-B′).

To examine the contribution of PCs during lamellar development and expansion, we used the *fgf10b:nEOS* reporter line. Digital segmentation of developing lamellae at different maturation stages in a 14 dpf larvae revealed the first steps of lamellae formation (Fig. 7D,E). In the least mature lamella, a few *fgf10b:nEOS*-positive cells could be visualized at the base of the lamella between the filament arteries (Fig. 7D, top, E). Notably, a cluster of nEOS-positive cells accumulated adjacent to the still-growing OMC (wide arrow in Fig. 7D top), suggesting a concurrent formation for both the OMC and PCs. Once the OMC tip cell reached the AFA, the OMC was fully formed and closed (Fig. 7D middle). Additional PCs emerged, as the lamella grew (Fig. 7D bottom, Fig. 7E). These observations suggest that lamellae form via a close interaction between a migrating endothelial tip cell and emerging PCs.

## Early gill development is reflected in the adult gill architecture

To gain a comprehensive understanding of gill filament and lamellae development, we quantified their number and spatial distribution across the four gill arches between 3.5 and 14.5 dpf. Consistent with our previous observations, there were more medial than lateral filaments at 3.5 dpf in all four arches (Fig. 8A,B), and a comparison across arches confirmed the maturation gradient of gill filament and lamellar development (Fig. 8B,C,E,F,H,I,K,L). The quantification further confirmed a slight temporal lag at the onset of gill filament formation, with medial filaments always appearing before the corresponding lateral filament within a holobranch, across all four arches (Fig. 3C, Fig. 4C, Fig. 8B,E). This pattern aligned closely with our timelapse observations from the early 3 dpf embryo (Fig. 3). At 5.5 dpf the trend continued, highlighting the significant developmental delay of lateral filaments previously observed (Fig. 8D,E, Fig. 4B,C). Accordingly, the first lamellae were observed on the more mature medial filaments and in the more mature arches (Fig. 8D,F). The difference in the number of lateral versus medial filaments was less strong by 10.5 dpf and had disappeared by 14.5 dpf (Fig. 8G,H,J,K). From 10.5 dpf onward, lamellae were present in all four arches, and the difference in number of lamellae on lateral versus medial filaments remained strikingly evident at 14.5 dpf (Fig. 8I,L). For medial filaments, the number of lamellae (and thus the filament length) was highest approximately midway to two-thirds along the dorso-ventral axis for all four arches, consistent with our earlier observation that this is where medial filaments first emerge (Fig. 8J,L, Fig. 3). For lateral filaments, the peak was slightly shifted ventrally, consistent with our observation that, in the ventral region, lateral filaments emerge

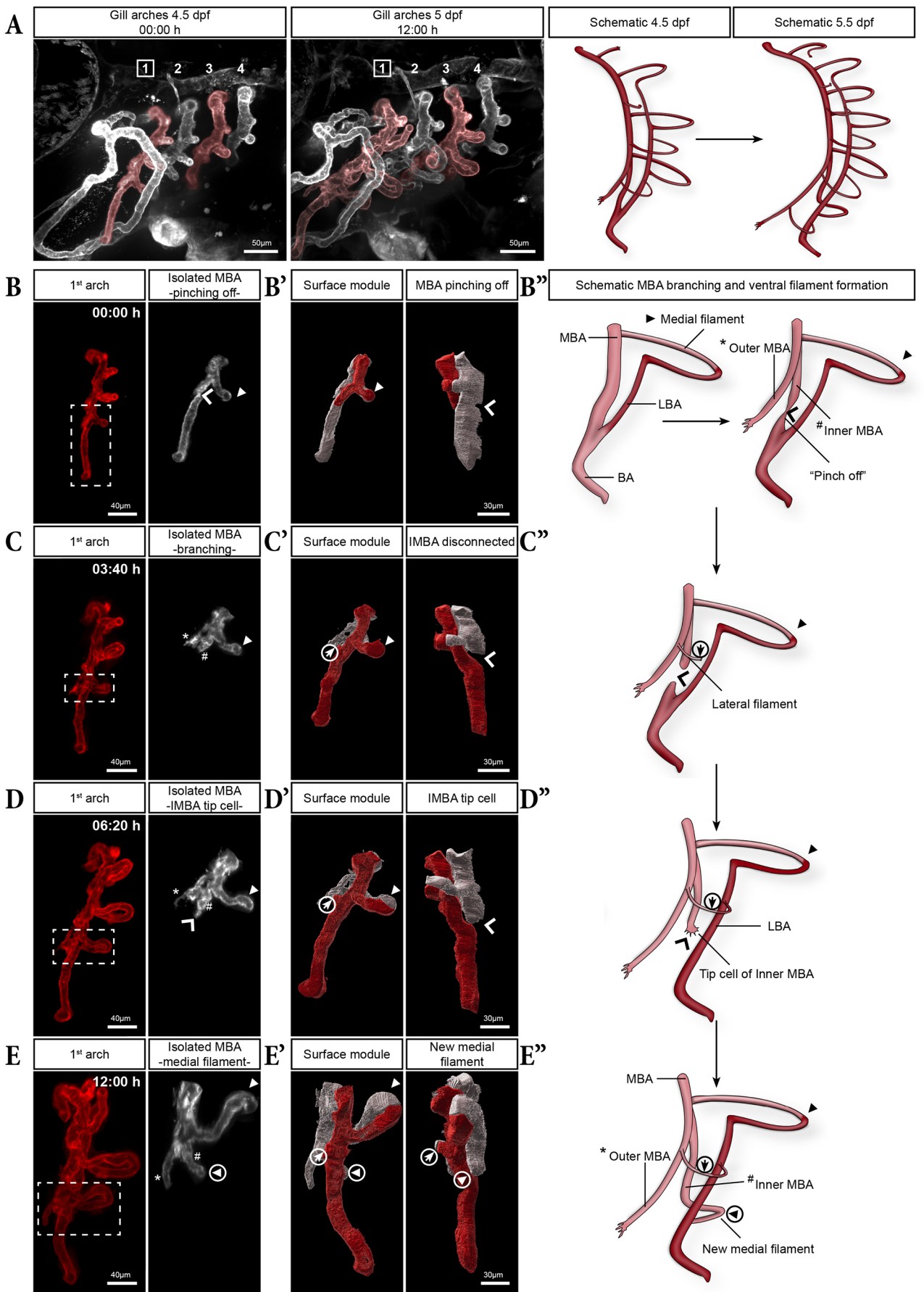

**Fig. 5.** See next page for legend.

**Fig. 5. MBA branching and formation of filaments in the ventral region of the gill arches.** (A) Maximum intensity projection (MIP) from a 12 h timelapse of the gill vasculature from 4.5 to 5 dpf, with corresponding schematics of the first gill arch. (B-E″) Snapshots of the timelapse at the indicated times, showing pinching and subsequent detachment of the medial branchial artery (MBA) (dorsal) from the branchial artery (BA) (ventral) (wide arrowheads in B-D″) and the emergence of the first ventrally formed medial and lateral filaments (encircled arrowheads and arrows) (C-E″). For each time point, the left panel displays the digitally isolated first gill arch and the right shows a magnified view of the ventral MBA region only (pseudo-coloured in white, corresponding to the dashed box in the left panel). Panels B′,C′,D′ and E′ display a surface rendering of the same region and a rotated view, to highlight the 'pinch-off' event (wide arrowheads). Surface modules are colour-coded to differentiate between the medial (white) and lateral (red) parts of the arch. Panels B″,C″,D″ and E″ show the corresponding schematics. Panel C shows bifurcation of the MBA into the inner (IMBA; #) and outer (OMBA; *) medial branchial arteries.

before medial filaments (Fig. 8A, Figs 5 and 6). Notably, the medial filaments (grey curve) were longer (had more lamellae) than the lateral ones (red curve) over most of the dorso-ventral axis, but became smaller in the most ventral part (crossing of the grey and red curves in Fig. 8L). This is consistent with our previous observations that medial filaments formed before lateral ones in the region dorsal to the MBA branching point (Fig. 4A,C) and, in contrast, lateral filaments formed before medial ones in the region ventral to the MBA branching point (Fig. 8L, Figs 5 and 6). Strikingly, this developmental pattern appears to reflect the spatial organization and length of filaments first observed in 3-month-old mature fish (Fig. 1B).

To confirm the relationship between early developmental patterns and adult gill architecture, we quantified filament number and length in the first gill arch of 6- and 14-month-old zebrafish. As observed at 14.5 dpf (Fig. 8K), the number of medial and lateral filaments was equal at these adult stages (Fig. 9A,B). Moreover, total filament number remained unchanged between 6 and 14 months, indicating that no additional filaments formed during this period (Fig. 9B′). Instead, the existing filaments increased significantly in length (Fig. 9C,D). Strikingly, the filament length distribution across the dorso-ventral axis closely mirrored the number of lamellae distribution observed at 14.5 dpf (Fig. 9C, Fig. 8L). This suggests that medio-lateral differences in filament length along the dorso-ventral axis are established early and persist into adulthood. We propose that this asymmetry is the consequence of medial filaments forming before the lateral ones dorsal to the MBA branching point, whereas lateral filaments form first ventral to this branching point.

### Lamellar size differences reflect medio-lateral filament asymmetry

Having established that the gill architecture is determined early in development and maintained into adulthood, we next asked whether the asymmetry in filament growth observed during development was accompanied by corresponding variations in the cellular architecture of the lamellae, the site for gas exchange. We therefore quantified PC number and distribution in lamellae of adult zebrafish to assess potential functional implications of developmental asymmetries. To this end, the first gill arch was arbitrarily divided into three roughly equal sections: dorsal, intermediate and ventral (Fig. 10A). Tissue clearing ensured comprehensive and holistic analysis of lamellae on both medial and lateral filaments along the dorso-ventral axis (Fig. 10B,C; Fig. S4; Movie 5).

In the dorsal third of the arch, medial filament lamellae contained more PCs than lateral filament lamellae (Fig. 10D). This difference diminished in the intermediate region and reversed in the ventral

third, where lamellae on lateral filaments contained more PCs than those on medial filaments (Fig. 10D). Notably, this medio-lateral asymmetry in PC number closely resembled the medio-lateral difference in filament lengths (Fig. 9C), with longer filaments in each subsection exhibiting lamellae containing more PCs (Fig. 10E). Accordingly, there was a positive correlation between the number of PCs per lamellae and the length of the corresponding filament in each section (Fig. 10E).

However, this correlation did not hold consistently when considering medial and lateral filaments separately. In medial filaments, the number of PCs per lamella was lowest in the ventral third, where the filaments are shortest, but surprisingly similar between the dorsal and intermediate sections, despite filaments significantly differing in lengths (Fig. 10F). Similarly, in lateral filaments the number of PCs per lamella was lowest in the dorsal region, where the filaments are the shortest, but was similar for filaments in the ventral and intermediate region, which differ in filament length (Fig. 10F).

Although a higher PC number suggests a larger lamella and potentially greater respiratory capacity, the spacing between individual PCs must also be considered to accurately determine lamellar size. In general, PCs were spaced further apart in lamellae containing more PCs (Fig. S5). Consequently, a greater number of PCs corresponds to larger lamellae, as reflected in the matching distributions of lamellar surface area (Fig. 10G) and the PC number distribution (red in Fig. 10F) along the dorso-ventral axis.

Overall, while there is a general correlation between filament length and PC number when comparing medial and lateral filaments (Fig. 10E), this relationship is not strictly linear within each group (Fig. 10F). This suggests that additional factors, such as filament identity or dorso-ventral position, may influence PC number and, therefore, lamellar size and respiratory capacity.

### DISCUSSION
### The formation of medial and lateral filaments follows a stereotyped pattern that underlies the architecture of the adult gills

In this study, we characterized the spatiotemporal development of the gills of zebrafish from their formation in the embryo to adulthood. While previous studies have provided detailed descriptions of the vascular anatomy of the developing zebrafish (Isogai et al., 2001) and of the mature gill (Olson, 2002), the precise developmental processes of the gill vasculature has remained unclear. Our data suggest that early gill development plays a pivotal role in shaping the mature gill architecture. One of our key findings is the difference in the temporal and spatial patterns of medial and lateral filament formation. This difference is apparent right from the onset of filament formation, where medial filaments emerge first, followed by the extension of the LBA, observed as early as 3 dpf (Fig. 3C). Lateral filaments emerge then from the MBA after the LBA formation and fuse with it.

Our findings indicate that this stereotyped, sequential pattern repeats for each filament dorsally. Further medial filaments consistently emerge first, contributing to the elongation of the LBA that is required for new lateral filaments to emerge and anastomose with the established LBA. Interestingly, lateral filaments are much smaller than their medial counterparts and initially lack the vascular loop seen in medial filaments. Besides the temporal difference, this structural difference may account for the length variation between the two filament types observed during development and in mature gills (Fig. 1B). We propose a model in which, dorsally, medial filaments

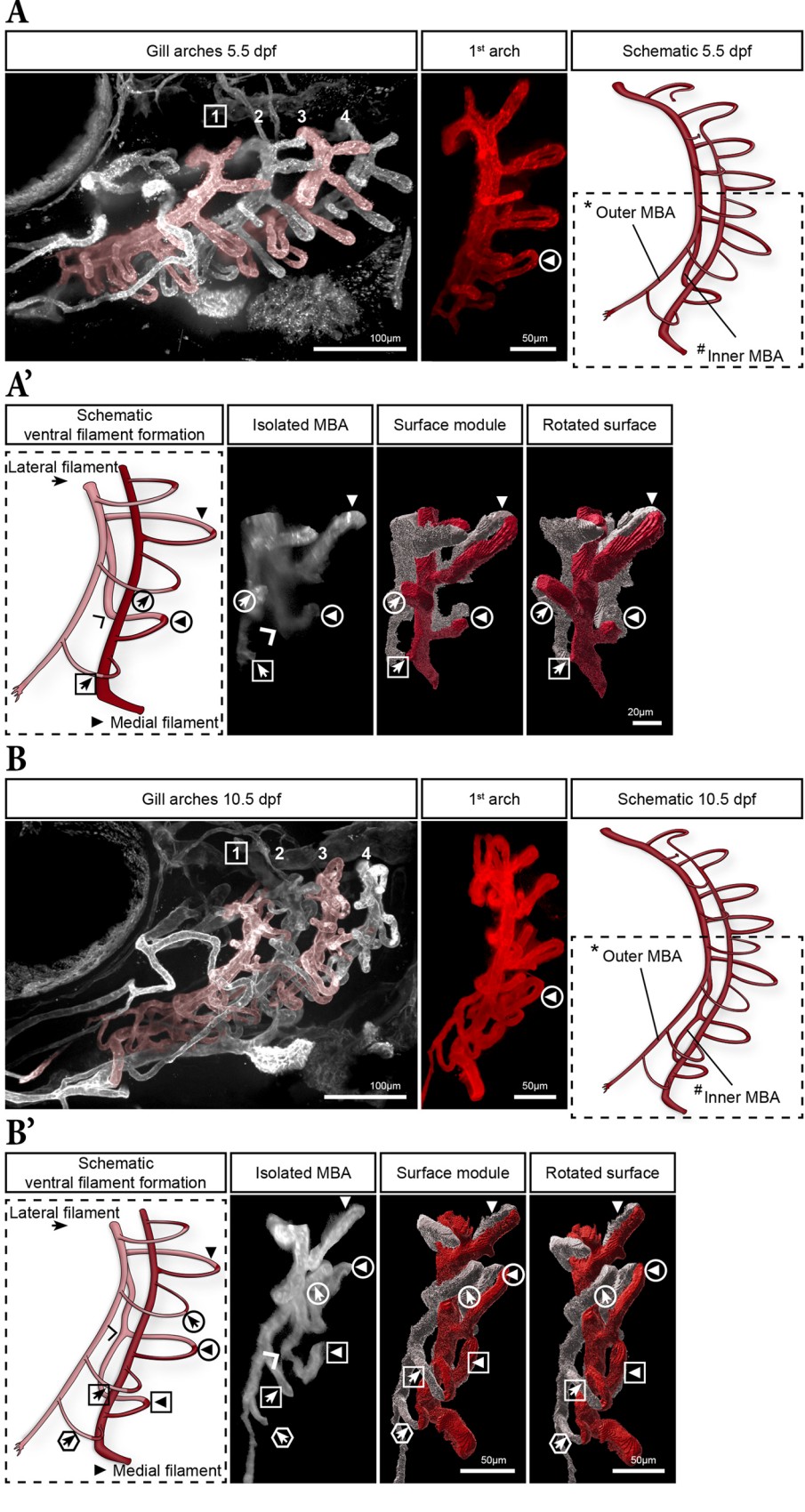

**Fig. 6. Ventral medial and lateral filament formation between 5 and 10 dpf.** (A) Overview of the gill vasculature in a 5.5 dpf larva (left) with the digitally isolated first gill arch (middle) and the corresponding schematic (right). (A′) Magnified, colour-coded schematic (dashed box in A) displaying the medial (white) and lateral (red) parts of the ventral area of the gill arch (left). The digitally isolated medial branchial artery (MBA) corresponds to the medial part of the scheme, and the two right panels are colour-coded surface renderings (medial in white, lateral in red) showing filament order and orientation in the ventral region of the arch. A new lateral filament formed ventrally (boxed arrow), and the first ventrally-formed medial and lateral filaments increased in size (encircled arrowhead/arrow). (B) Overview of the gill vasculature in a 10.5 dpf larva (left), with digitally isolated first gill arch (middle) and corresponding schematic (right). (B′) Magnified and colour-coded scheme (dashed box in B) showing the medial (white) and lateral (red) parts of the ventral area of the first arch (left). The digitally isolated MBA corresponds to the lower part of the scheme, and the two right panels are colour-coded surface renderings as in A′. New filaments formed (boxed arrowhead and hexagonal arrow) ventral from the lateral filament (boxed arrow).

provide the structural framework (the LBA) for the later development of lateral filaments (Fig. 4C). Although medial filaments were previously shown to form before lateral ones in another teleost species (Coughlan and Gloss, 1984), our findings provide the first explanation for this sequence by elucidating precisely the distinct developmental processes of each filament type in zebrafish.

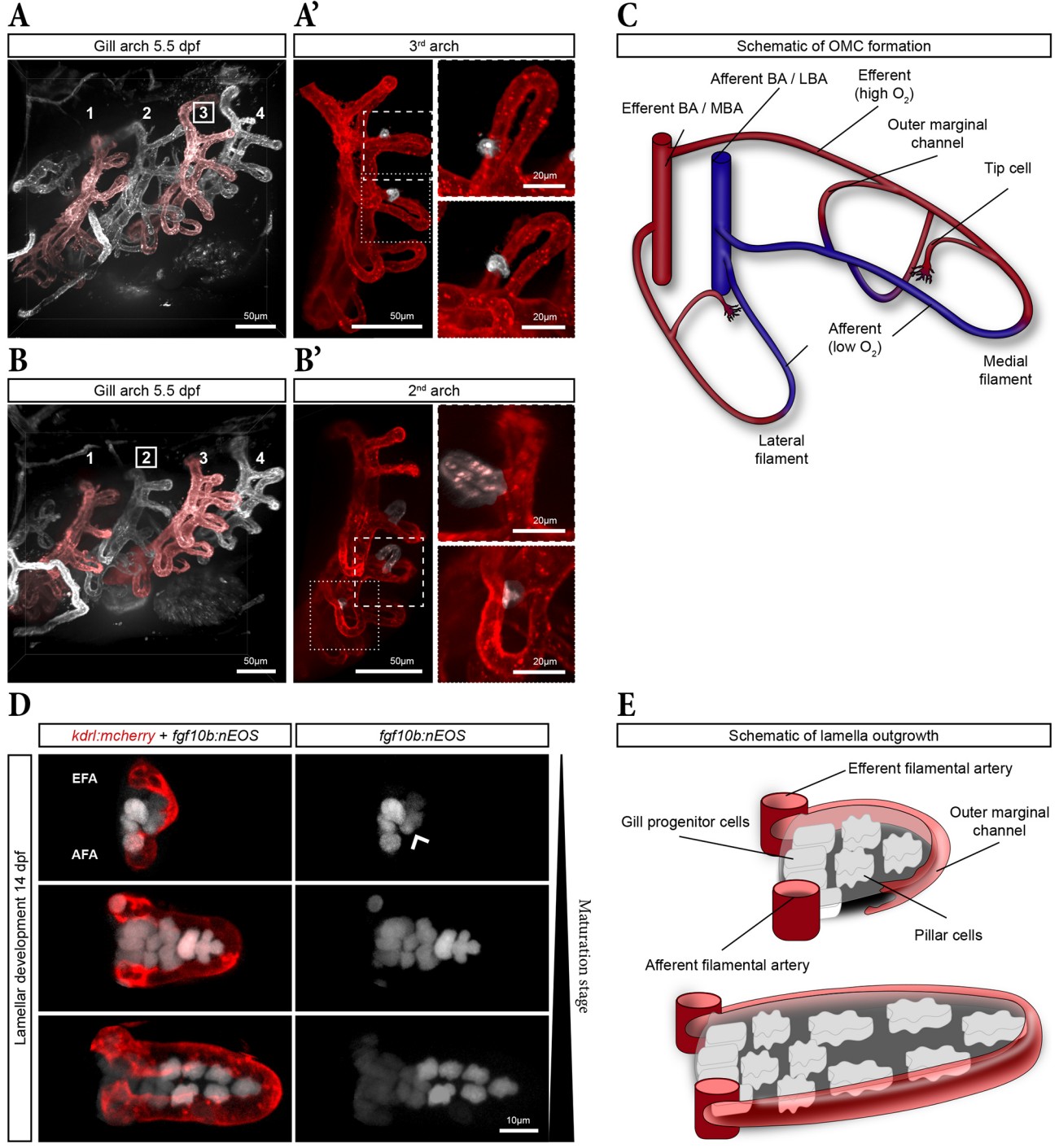

**Fig. 7. Interplay of OMC tip cells and pillar cells control lamellae morphology and outgrowth.** (A) Overview of the gill vasculature in a 5.5 dpf larva (left). (A′) Digitally isolated third gill arch with close-up views of two medial filaments (dotted and dashed boxes) revealing tip cells of the outer marginal channel (OMC; pseudo-coloured in white) in emerging lamellae. (B) Overview of the gill vasculature in a 5.5 dpf larva. (B′) Digitally isolated second gill arch with close-up views showing a fully closed OMC (pseudo-coloured in white, dashed box, upper panel) on a medial filament and a tip cell on a lateral filament (pseudo-coloured in white, dotted box, lower panel). (C) Schematic illustrating OMC formation. (D) Differently mature lamellae in a 14 dpf *kdrl:mCherry; fgf10b:nEOS* larvae. nEOS-positive pillar cells (white) are already in contact with the tip cell of the OMC (red) at onset of OMC formation (wide arrowhead). (E) Schematic describing the interplay of pillar cells and the OMC tip cell during lamellae outgrowth.

Moreover, we demonstrate that the formation of lateral and medial filaments differs significantly along the dorso-ventral axis of the gill arches. The branching of the MBA into the OMBA and IMBA, followed by the pinch-off of the IMBA from the BA, defines a distinct anatomical boundary between the dorsal and ventral regions. This bifurcation has also been reported in other adult teleost fish, suggesting that it may be a conserved morphological feature (Morgan and Tovell, 1973; Olson, 2002). However, its developmental dynamics and functional significance has not yet been addressed. We propose that this vascular remodelling is essential for directing deoxygenated blood from the heart exclusively into the LBA, thereby ensuring oxygenation as blood

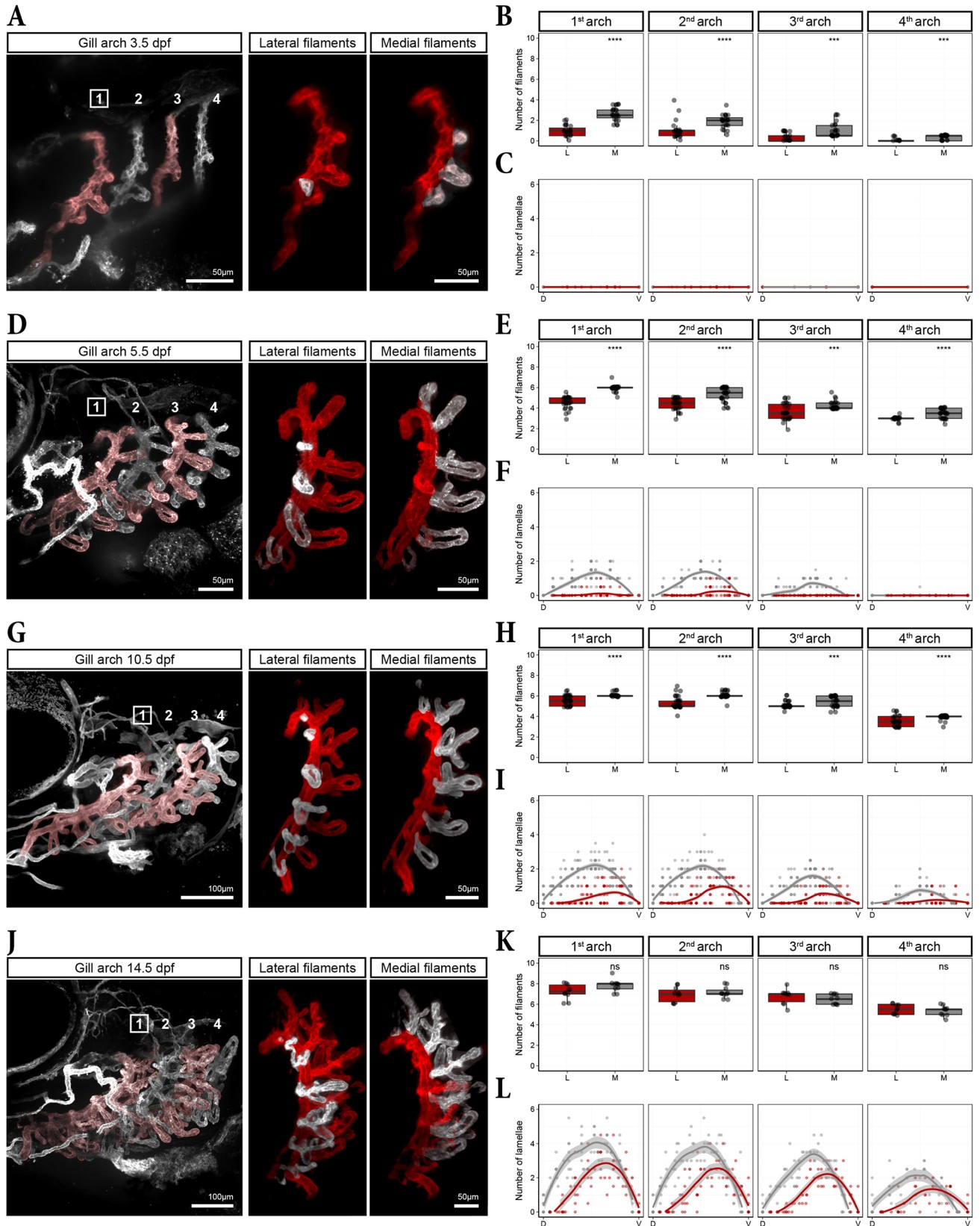

**Fig. 8. Spatio-temporal patterns of gill filament and lamellae formation.** (A,D,G,J) Overview of the gill vasculature at the indicated time points (left panel) and the first gill arch digitally isolated from the overview with lateral (middle panel) and medial (right panel) filaments highlighted in white. (B,E,H,K) Box plots showing the number of medial and lateral filaments for each arch. Box plot centre: median. Bounds: 25th and 75th percentiles. Whiskers: minimum/maximum values. Mann–Whitney *U*-test (two-tailed) was applied: ***P≤0.001; ****P≤0.0001; ns, not significant. (C,F,I,L) Plots displaying the number and distribution of lamellae per filament across the dorso-ventral axis of the gill arch. Curves represent fitted LOESS model with s.e. (grey area). (B,C) *n*=25, (E,F) *n*=31, (H,I) *n*=35, (K,L) *n*=10.

passes through the filaments and lamellae before reaching the MBA. Additionally, we observed that in the ventral region, lateral filaments, deriving from the OMBA, form first, followed by medial filaments originating from the IMBA. This contrasts sharply with the dorsal region, where medial filaments appear before lateral ones, and likely accounts for the slight ventral shift in the filament length peak for lateral versus medial filaments (Fig. 8I,L). These spatiotemporal developmental differences presumably contribute to the length disparity observed between the two filament types in the dorsal and in the ventral region, which remains in the mature gills (Fig. 9A,B).

The MBA bifurcation marks a clear difference between dorsal and ventral development, which initially explains why medial filaments are longer dorsally, while the pattern is reversed ventrally. Observing a similar pattern in adult fish, with an inversion in the filament length roughly located in the region of the bifurcation (Fig. 9C; Fig. S2), strongly supports the hypothesis that the developmental pattern accounts for the asymmetry in filament lengths across all gill arches in adult.

In addition to the intrinsic developmental programme, internal blood pressure (Gebala et al., 2016; Ghaffari et al., 2015) and external mechanical forces, such as hydrostatic pressure generated

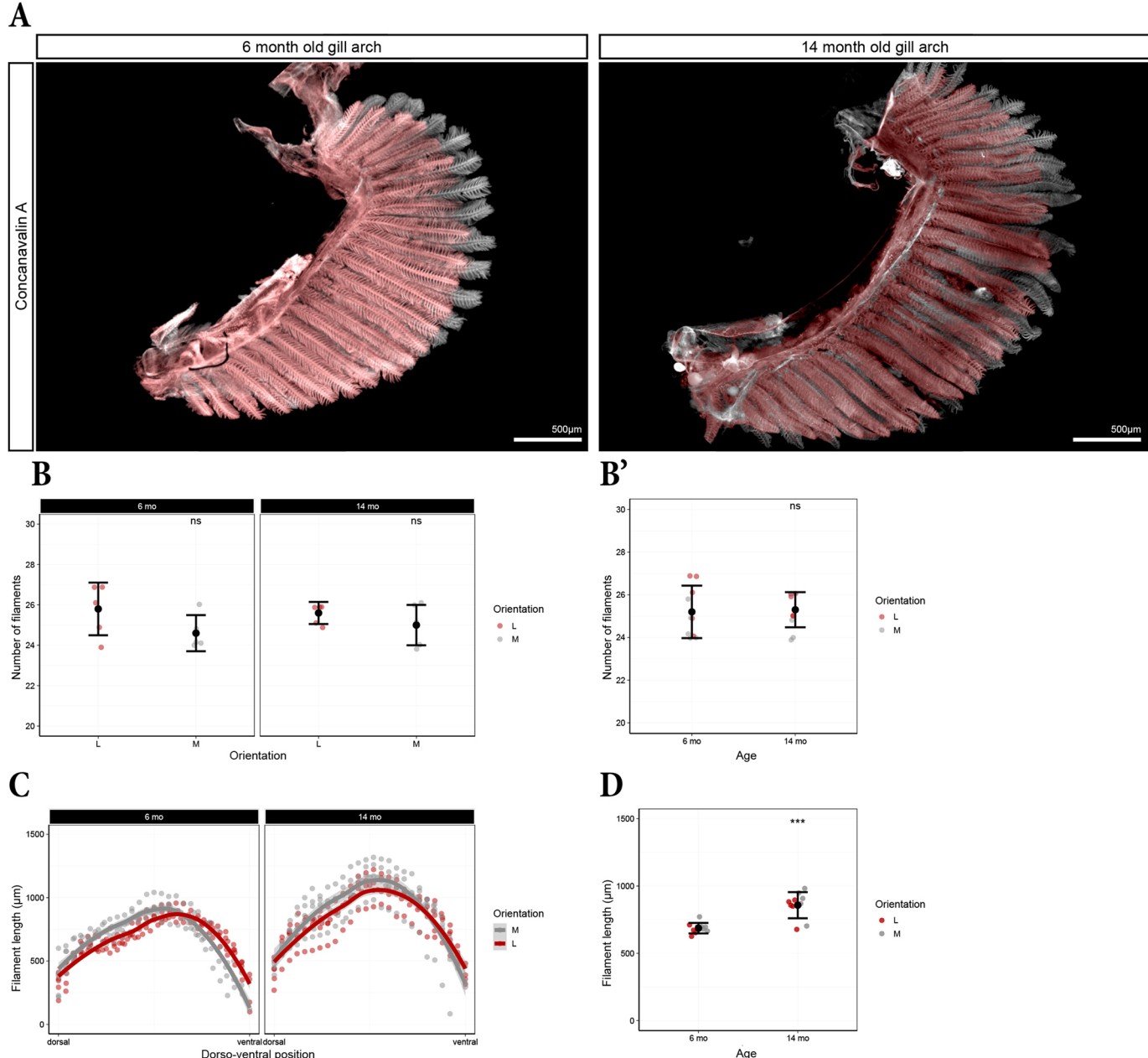

**Fig. 9. Filaments still grow in adult zebrafish, but their number does not increase.** (A) First gill arch from 6- and 14-month-old fish cleared with ECi and stained with Concanavalin A. Medial filaments are pseudo-coloured in white and lateral filaments in red. (B,B′) Number of medial and lateral filaments in 6- and 14-month-old fish comparing medial versus lateral (B) and 6 versus 14 months (B′). One point represents one hemibranch (first arch). (C) Distribution of the medial and lateral filament length in the first gill arch along the dorso-ventral axis. Curves represent fitted LOESS model with s.e. (grey area). Each point represents a single filament. (D) Length of medial and lateral filaments in 6- versus 14-month-old zebrafish. Mann–Whitney *U*-test (two-tailed) was applied. \*\*\**P*≤0.001; ns, not significant. Error bars indicate mean value±s.d. (B-D) *n*=5.

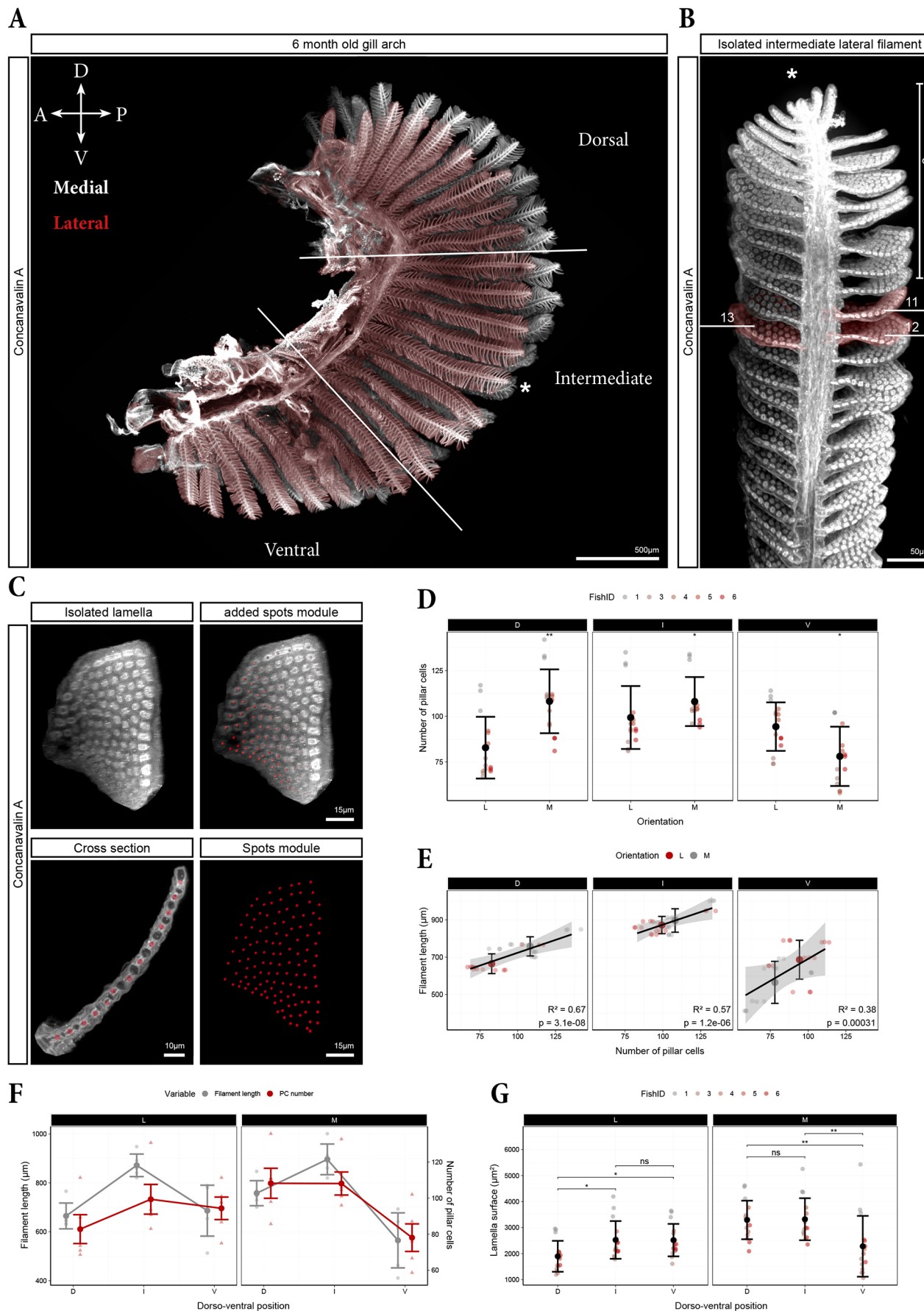

**Fig. 10.** See next page for legend.

**Fig. 10. Relation between lamellae and filament size along the arch.**
(A) The gill arch of 6-month-old fish was cleared and stained with Concanavalin A. After imaging, it was arbitrarily divided into three thirds: dorsal (D), intermediate (I) and ventral (V). An individual filament (asterisk) serves as a continuous example for the quantification steps. (B) Three consecutive lamellae (pseudo-coloured in red) were digitally isolated, beginning with the 11th lamella. (C) Digitally isolated lamellae (left) analysed with Imaris's spots module, recognizing individual pillar cells (PCs) and marking them as red dots. The cross section (bottom left) demonstrates the specificity of the applied spots module. (D) Comparison of the number of PCs per lamellae in medial (M) versus lateral (L) filaments in the divided thirds of the arch. (E) Plot showing the relation between the number of PCs per lamellae and the filament length for lateral and medial filaments in the three different regions. (F) Same data as in E, showing the direct comparison of PC number per lamella (red) with corresponding filament length (grey) in lateral (left) and medial (right) filaments. (G) Direct comparison of the lamella surface of lateral (left) and medial (right) filaments in the three regions on the arch. (D,G) Data points represent individual lamellae and are colour-coded per fish (1 to 6). Three lamellae for each region in each animal, $n=5$. Error bars indicate mean value±s.d. Mann–Whitney $U$-test (two-tailed) was applied: *$P \leq 0.05$; **$P \leq 0.01$; ns, not significant. (E,F) Each small point represents one fish (mean of three lamellae), error bars indicate mean value±s.d.

by water flow through the buccal cavity, might influence gill filament formation and angiogenesis (Chugh et al., 2022). Furthermore, forces, such as blood flow, muscle contraction or tissue stiffness, are mechanical cues that can be transduced at the molecular level to regulate gene expression (Panciera et al., 2017). Recent studies showed that Aquaporin-mediated water inflow and resulting hydrostatic pressure can influence endothelial tip cell migration and protrusion (Kondrychyn et al., 2025). We assume that these factors, in combination with the developmental programme, could contribute to both the elongation and spatial organization of gill filaments, ultimately shaping the architecture of the gill arches. Our description of the process paves the way to use the zebrafish gill to further address the influence of blood pressure on vascular development *in vivo*.

### Development, size and distribution of lamellae
Our observations indicate that lamellar development is initiated by the migration of an endothelial tip cell from the EFA towards the AFA, establishing the OMC. This aligns with an older study suggesting that ECs play a central role in the onset of gill development (Morgan, 1974). Using the *Tg(fgf10b:nEOS)* reporter line, we found that PCs accumulate adjacent to the OMC endothelial tip cell, suggesting that they contribute to lamellar outgrowth. PCs have been reported to arise initially from a stem cell niche of cranial neural crest cells at the filament tip (Fabian et al., 2022; Mongera et al., 2013; Stolper et al., 2019) and to be maintained by homeostatic stem cells (SCs) at the base of the lamella, at least in medaka (Stolper et al., 2019). An important avenue of exploration will be to investigate how PCs and ECs interact during lamellae outgrowth and how they influence each other, potentially shedding light on fundamental principles of vascular development.

We show that smaller lamellae contained fewer, more densely arranged PCs, particularly in the early stages of lamellar outgrowth, but gradually became more spaced as the lamellae expanded. Consistent with the role of mechanical forces proposed for filament development, blood flow may also contribute to expand the vascular space, increasing the distance between neighbouring PCs and effectively unfolding the lamellae from within, resembling lumen formation (Chugh et al., 2022).

Interestingly, we uncovered not only an asymmetry in filament length between medial and lateral filaments, but also a corresponding asymmetry in lamella size across the dorso-ventral axis. While PC

number generally correlated with filament length, this relationship did not hold consistently within medial or lateral filaments. We hypothesize that the observed structural asymmetry correlates with functional roles of filaments and lamellae in optimizing gas exchange. Longer filaments and larger lamellae, concentrated at the dorso-intermediate region, likely experience the highest water flow across the gill surface. These regions may be optimized for gas exchange. In contrast, dorsal lateral filaments and those in the ventral region may play subordinate roles, potentially modulating flow rather than maximizing gas exchange.

### Lifelong growth and maturation of the gills
Extending beyond early development, our study provides the first detailed characterization of gill formation across the entire lifespan of zebrafish. While we initially hypothesized that new filaments continuously arise throughout adulthood, as previously described in medaka (Stolper et al., 2019), our observations revealed otherwise. Although zebrafish continue to grow throughout life (Singleman and Holtzman, 2014), the number of filaments per arch stabilises at around 25 by ~6 months of age, consistent with previous quantifications (Ramel et al., 2021 preprint). Rather than generating new filaments, lifelong gill growth primarily occurs through elongation of existing filaments via the addition of new lamellae. The presence of SCs in the growth domain at the distal tip of the filament in zebrafish, like in medaka (Fabian et al., 2022; Stolper et al., 2019), supports the idea of continuous filament growth. This further suggests that the longest filament is also most likely the oldest. Notably, we observed a row of densely packed, less differentiated PCs at the base of growing lamellae, reminiscent of the homeostatic SCs described in medaka (Stolper et al., 2019). These progenitors may contribute to the lifelong growth of lamellae to meet elevated oxygen demands and to adapt to environmental changes, such as hypoxia, temperature or injury (Chou et al., 2008; Jonz, 2024; Tzaneva et al., 2011).

With their high physiological relevance and structural analogy to the mammalian lung, gills provide a unique platform to study diverse cell biological processes, many of which can be visualized dynamically and *in vivo* through live imaging. With this study, we establish the zebrafish gills as an additional model to investigate complex biological processes, including the mechanisms underlying three-dimensional angiogenesis and the influence of blood and hydrostatic pressure, the contribution of neural crest-derived stem cells to the development and regeneration of filaments and lamellae, and branching morphogenesis involving multiple cell types. Our findings provide a precise and quantitative reference framework for this valuable model and pave the way for future functional studies.

## MATERIALS AND METHODS
### Zebrafish lines and maintenance
Procedures involving animals were conducted according to the guidelines of the Goethe University Frankfurt and the German Animal Welfare Act, and approved by the German authorities (veterinary department of the Regional Board of Darmstadt). Zebrafish (*Danio rerio*) were raised and maintained under standard conditions (Kimmel et al., 1995). Transgenic lines *Tg(kdrl: Hras.HRAS-mCherry)*$^{s896}$ (*kdrl:mCherry*) and *Tg2(NLS-Eos)*$^{el865}$ (*fgf10b: nEOS*) have been described previously (Chi et al., 2008; Fabian et al., 2022).

### Spinning disc microscopy on fixed samples
Fish were euthanised by hypothermic shock in 4°C cold system water. Larvae or dissected gills were then fixed in 4% paraformaldehyde (PFA) in PBS for 2 h at room temperature (RT) or overnight at 4°C. The samples were then washed three times for 10 min in 1× PBS. Larvae were mounted in 0.3% low melting agarose in E3 in glass-bottom Petri dishes (Ibidi, 35 mm).

All fluorescent images were taken using a fully motorized Nikon-Ti spinning disc microscope with NIS 5.0 software.

Fixed gills were washed for 30 min at 4°C in ice cold 1× PBS with 0.3% Triton X-100 to enhance tissue penetration and promote the substitution of water inside the tissue with high refractive index (RI) medium. Afterwards the samples were incubated in 10% normal goat serum (NGS) for 2 h at RT. Then 5 mg/ml Concanavalin A (Thermo Fisher Scientific, C21421) was diluted 1:500 in 2% NGS and incubated for ~60 h at 4°C. After incubation, the specimen was washed in 1× PBS overnight.

### Ethyl cinnamate-based clearing procedure
The samples labelled with Concanavalin A were transferred into the bracket of the Leica autostainer (TP1020). The specimen was placed into an ascending ethanol series, increasing from 50% to 70%, and finishing with 2× 100% ethanol, to slowly dehydrate the tissue. Each dehydration step was performed for 4 h. Following 2 h of incubation in ethyl-cinnamate (ECi) (Sigma-Aldrich, 112372) the sample was either directly imaged or stored in glass vessels at RT for several weeks without losing its signal (Klingberg et al., 2017). Using fine forceps, gill arches were separated and placed on a glass slide covered in ECi and sealed with a cover slip.

### CUBIC-based clearing procedure
Fixed gills were washed three times for 30 min at 4°C in ice-cold 1× PBS containing 0.3% Triton X-100 and incubated overnight at 37°C in CUBIC Reagent-2 (Susaki et al., 2015). Using fine forceps, gill arches were then separated and mounted on a glass slide in Reagent-2 and sealed with a coverslip.

### Timelapse microscopy
Live embryos until 5 dpf were anesthetized using 50 µg/ml Tricaine (1:75) and mounted in 0.3% low melting agarose in E3 in glass-bottom Petri dishes (Ibidi, 35 mm). Timelapse movies were acquired on a fully motorized Nikon-Ti spinning disc microscope with NIS 5.0 software for 15 h. Z-stacks were captured at 1 µm intervals every 25 min and subsequently analysed using Imaris image analysis software.

### Electron microscopy
Dissected gills were fixed in 2.5% glutaraldehyde and 4% paraformaldehyde in 0.1 M cacodylate buffer (pH 7.2) for 2 h at RT. For TEM, samples were washed twice in 0.1 M cacodylate buffer containing 2% sucrose, post-fixed in 1% reduced osmium tetroxide, dehydrated and embedded in Araldite resin. Ultrathin sections (50 nm) were cut using an ultramicrotome (Leica), and ribbons of sections were mounted on Formvar-coated copper slot grids. Contrast was enhanced with 5% uranyl acetate in methanol/water followed by lead citrate. Micrographs were acquired using a Zeiss TEM 900 transmission electron microscope operated at 80 keV in brightfield mode and equipped with a Tröndle 2K camera.

For SEM the samples were critical point dried after fixation, followed by gold sputter coating using the BALTEC MED 020 coating system. Imaging was performed using a Hitachi S-4500 at 10-15 kV.

### Isolation of gill arches and pillar cell quantification
Imaris software (Oxford Instruments, v.10.1.1) was used for visualization, image processing, quantification and video rendering of three-dimensional images. Digital isolation of gill arches, filaments and lamellae was performed manually through Imaris surface module. The surface module was used to highlight filaments and endothelial tip cells.

The spots module was used to quantify the number of PCs per lamellae on ECi-cleared gill samples stained with Concanavalin A (Kato et al., 2007). For each filament, three lamellae located beneath the growth domain at the distal filament tip were quantified. The first ten lamellae, starting from the distal tip of each filament, were considered to be part of the growth domain (Stolper et al., 2019) and were therefore excluded from the analysis. Spots with a diameter of 4.50 µm were applied and manually corrected and validated if necessary. Ultimately, the module provided information such as the total number of PCs and the distance to nearest neighbour. All individual steps, including digital isolation, surface rendering and PC quantification,

are shown in a single compiled video that summarizes the entire workflow as shown in Fig. 10A-C (Movie 5).

### Movie creation
Movies were created using the key frame animation mode of Imaris software (version 10.1.1) with a frame rate of 5 (Movies 2, 3) and 24 (Movies 1, 4, 5) frames/sec.

### Quantification of filament number and lamellar distribution
Individual gill arches were segmented using the surface module in Imaris. Each arch was manually delineated across successive z-planes to generate a three-dimensional surface corresponding to a single arch. The relevant fluorescent channel was subsequently masked to ensure that only the defined surface was visible. The number of lamellae and filaments was then manually quantified and annotated based on their position along the dorso-ventral axis for each of the four arches. Fully matured lamellae and filaments were counted as one unit, while those still under development were counted as 0.5 unit. For 6- and 14-month-old zebrafish, only the first gill arch was analyzed. Filament orientation, length and position along dorso-ventral axis were quantified by hand. To enable comparison across arches with varying filament numbers, filament positions within each arch were minimum-maximum scaled to a [0,1] range. The distribution of filament length (6 and 14 months) or lamellae number (3, 5, 10 and 14 dpf) along the scaled dorso-ventral axis (0=dorsal, 1=ventral) was visualized using a locally estimated scatterplot smoothing (LOESS) curve.

### Plotting and statistical analysis
All analyses were conducted in R (v.4.3.1). Data visualization used ggplot2 (v.3.5.2), with combined plots arranged with patchwork (v.1.3.0). Data wrangling was performed using dplyr (v.1.1.3) and tidyverse (v.2.0.0). Statistical analyses employed ggpubr (v.0.6.0) and rstatix (v.0.7.2). Unless otherwise stated, statistical comparisons were performed using the Wilcoxon rank-sum test: For comparison of two groups, the stat_compare_means function was used (ggpubr v.0.6.0). For comparisons across multiple groups, pairwise Wilcoxon rank-sum tests were performed using the pairwise_wilcox_test function with $P$-values adjusted using the Holm method (v.0.7.2.). These non-parametric tests were chosen for their robustness to non-normal distributions, assuming similar distribution shapes between groups.

### Lamellae surface area
Because lamellae consist of a single layer of PCs, their surface area was approximated as:

$$A_{Lamella} = N \times A_{Cell},$$

where $N$ is the number of cells per lamella. Assuming biologically efficient hexagonal arrangement of cells, the area occupied by one PC, $A_{Cell}$, was estimated as:

$$A_{Cell} = \frac{\sqrt{3}}{2} \times (DTNN)^2,$$

with DTNN representing the distance to nearest neighbour. The total lamella surface was therefore calculated as:

$$A_{Lamella} = N \times \frac{\sqrt{3}}{2} \times (DTNN)^2.$$

This geometric approximation assumes regular two-dimensional packing with isotropic spacing.

### Acknowledgements
We thank Dr Peter Fabian for providing the *fgf10b:nEOS* transgenic line and Dr Julien Rességuier for valuable insight into gill biology and methodology. We are also grateful to Dr Maik Bischoff for the engaging exchange. We are very grateful to Prof. Stefan Eimer for the access to the Imaris software and to Prof. Ivica Grgic for providing the tissue clearing setup. Special thanks go to the animal care team for their excellent maintenance of the fish facility, and to M. Kamprad and M. Heyde for their valuable technical assistance.

**Competing interests**

The authors declare no competing or financial interests.

**Author contributions**

Conceptualization: M.P., V.L.; Funding acquisition: V.L.; Investigation: M.P., M.B.; Methodology: M.P., M.B.; Project administration: M.P., V.L.; Supervision: V.L.; Formal analysis: M.P., A.M.; Visualization: M.P.; Writing – original draft: V.L., M.P.; Writing – review & editing: M.P., V.L., A.M.

**Funding**

This work was supported by Clusterproject EnABLE, which is funded by the Hessisches Ministerium für Wissenschaft und Kunst (V.L.). Open Access funding provided by Goethe University Frankfurt. Deposited in PMC for immediate release.

**Data and resource availability**

All relevant data can be found within the article and its supplementary information.

**Peer review history**

The peer review history is available online at https://journals.biologists.com/dev/lookup/doi/10.1242/dev.204984.reviewer-comments.pdf

**Special Issue**

This article is part of the Special Issue 'Lifelong Development: the Maintenance, Regeneration and Plasticity of Tissues', edited by Meritxell Huch and Mansi Srivastava. See related articles at https://journals.biologists.com/dev/issue/152/20.

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
