## [Peer Review File · Development (Cambridge, England)]

A developmental atlas of zebrafish gills links early vascular patterning to adult architecture

Mathieu Preußner, Anna Mertens, Marion Basoglu and Virginie Lecaudey
DOI: 10.1242/dev.204984

Editor: Steve Wilson

Review timeline

Original submission:	30 May 2025
Editorial decision:	26 June 2025
First revision received:	22 July 2025
Accepted:	25 July 2025

Original submission

First decision letter

MS ID#: dev.204984

MS TITLE: Lifelong development of zebrafish gills: Asymmetries in early vascular patterning predict adult gill architecture

AUTHORS: Virginie Lecaudey; Mathieu Preussner; Anna Mertens; Marion Basoglu

Dear Virginie,

I have now received all the referees reports on the above manuscript, and have reached a decision. The referees' comments are appended below, or you can access them online: please go to.

The overall evaluation is positive and we would like to publish a revised manuscript in Development after you have addressed the referees' comments. Please attend to all of the reviewers' comments in your revised manuscript and detail them in your point-by-point response. If you do not agree with any of their criticisms or suggestions explain clearly why this is so. Please send us a point-by-point response indicating your plans for addressing the referees' comments, and we will look over this and provide further guidance.

Reviewer 1

Advance summary and potential significance to field

Gills are complex structures essential for respiration in fishes. As fishes represent the largest group of vertebrates, understanding the development of gills is therefore highly important. While older studies have characterized the cellular components and anatomical maturation of gills, using largely histological and electron microscopy, how the complex vascular network of gills develops from embryonic to adult stages has remained a mystery. In the current study, the authors use zebrafish transgenic lines that label the endothelial and pillar cells of the gills to connect embryonic induction of vascular structures to heterogeneity of the adult vasculature that is likely important for maximizing respiration. A strength is the combination of time-lapse microscopy in larvae and tissue-clearing protocols in adults to show that adult anatomy reflects the timing of how different vascular structures first form in the embryo. This is a tour-de-force of microscopy and the

images shown are exquisite. They also do a wonderful job of schematizing and explaining what is very complicated anatomical development. While it is true that the study is largely descriptive, it represents what should become a definitive reference atlas for gill development in the genetically tractable zebrafish.

Comments for the author

I only have a few minor comments to improve this otherwise excellent manuscript.

1. The title is too long and should not contain a colon.
2. Several time-lapses are mentioned but only one Movie is included (the second movie is an overview of gill morphology). It would be helpful to include movies corresponding to all the time-lapses shown in the figures.
3. Lines 154-156, "Indeed, the immediately adjacent dorsal medial filament consistently corresponded to the very first-formed filament (arrowhead "1" in Fig. 2A), and could still be identified in mature fish (Fig. S2)." - I could not find the "arrowhead 1" in Fig. 2A and there were no annotations in Fig. S2 to indicate the very first-formed filament.
4. Is Tg2(NLS-Eos) a typo? Even though it is listed as such on ZFIN, this name does not make sense. The original paper called this fgf10b:nlsEOS.

Reviewer 2

Advance summary and potential significance to field

The article by Preussner et al. presents an excellent account of the processes involved in development of the gill vasculature in the model vertebrate, the zebrafish. This is a descriptive and complete account of gill development, and is the most detailed in this species. I find that the images provided are stunning and of unusually high quality and clarity. The supplemental movies were a nice addition as well. The only issue is that the authors make sweeping or inaccurate statements that overestimate the novelty of their work. For example, the medial vs. lateral asymmetry in gill development (or adults) is not novel and was reported in another species in 1984; previous studies on gill development, and gill vasculature, in zebrafish were not mentioned nor cited; and the zebrafish gill has been a well established model in biology for decades. I discuss these issues below in my specific comments. Presuming that the authors missed these important papers, I also provide citations. I also find the title, "Lifelong development of zebrafish gills" misleading since it is very well known that gills grow throughout life in fish. The present article has many strengths, particularly regarding the impacts of asymmetry on gill development, the chronology of anatomical changes and angiogenesis, but revisions are necessary to clarify what is novel about this paper.

Comments for the author

Lines 28-29. Kimmel et al. (1995) provided the first details on gill development in zebrafish and should be added to this list of citations. (The paper is already cited in the methods section for its procedures.)

Lines 40-41. The statement "and discouraging new investigators" is subjective and should be deleted.

Lines 67-68. Did the authors actually observe blood flow? If not, I suggest rephrasing this statement.

Line 69. A word is missing. "Blood"?

Figure 1H. Upper left, should be "Gill raker". Bottom panel should be titled with the singular, "Lamella cross section". Same for Fig. 7E.

Line 83-84. This statement is incorrect and must be revised: "uncovered a previously unrecognized, consistent medio-lateral asymmetry in filament length across all arches". This asymmetry is well-

known to occur in the gills of nearly all fish species. Moreover, the medial vs. lateral asymmetry in developing gill filaments, and appearance of lamellae etc., has already been observed in another species: see Coughlan and Gloss (1984). Early morphological development of gills in smallmouth bass (*Micropterus dolomieu*). *Canadian Journal of Zoology*. 62: 951-958. I suggest rephrasing the statement with a focus on the evidence that the authors provide that shows the impacts of this asymmetry on gill development.

Line 101. Please label the tip cell in Fig. 3A (first panel). It is not clear.

Line 156. There is no arrowhead in Fig. 2A.

Lines 190-191. It is stated, "First signs of developing lamellae were observed at 4 dpf", but the figure shows lamellae starting to develop in 5.5 dpf larvae. What were these first signs, and can an image at 4 dpf be provided, at least as a supplemental figure?

Lines 291-293. Regarding the statement: "While previous studies have provided detailed descriptions of the vascular anatomy of the developing zebrafish (Isogai et al., 2001), the description of the gill vasculature has remained sparse." Can the authors qualify "sparse" and perhaps cite these studies? At least one study has reported some aspects of gill vasculature in zebrafish: Jonz and Nurse (2003). *J Comp Neurol*. 461:1-17.

Lines 391-392. The first part of this sentence is not accurate and must be revised: "With this study, *we establish the zebrafish gills as a new and valuable model to investigate complex biological processes*...". Also, line 396 "this new model". The zebrafish gill has been an important model in biology for decades. The authors will need to correct these statements.

Line 459. Delete "several".

First revision

Author response to reviewers' comments

Point by point response to the reviewers

Dear Steve,

Thank you for considering our article (initially) entitled "**Lifelong development of zebrafish gills: Asymmetries in early vascular patterning predict adult gill architecture**" for publication in *Development's* current special issue: Lifelong Development: the Maintenance, Regeneration and Plasticity of Tissues. We are very grateful for the positive evaluation and the very insightful and constructive suggestions of the reviewers. We answered each of the reviewer's point and hope our answers are up to their expectations.

Reviewer 1 suggestions to the authors:

1. The title is too long and should not contain a colon.

We agree and changed the title accordingly to: A Developmental atlas of zebrafish gills links early vascular patterning to adult architecture.

2. Several time-lapses are mentioned but only one Movie is included (the second movie is an overview of gill morphology). It would be helpful to include movies corresponding to all the time-lapses shown in the figures.

We agree with the reviewer and have now added movies corresponding to the timelapses of Fig. 3A (Movie 1; Line: 105); Fig. 4B (Movie 3; Line: 132); Fig. 5 (Movie 4; Lines: 143, 161).

3. Lines 154-156, *"Indeed, the immediately adjacent dorsal medial filament consistently corresponded to the very first-formed filament (arrowhead "1" in Fig. 2A), and could still be identified in mature fish (Fig. S2)." - I could not find the "arrowhead 1" in Fig. 2A and there were no annotations in Fig. S2 to indicate the very first-formed filament.*

Thank you for pointing this out. This was an error on our part – the correct reference should have been to **Fig. 3A**, not Fig. 2A. We have updated the text accordingly. In Fig. 3A, an arrowhead clearly indicates the very first-formed filament (Line: 153). In Fig. S2 the first formed filament was pseudocoloured in white to highlight it. This was now added to the text and is stated in the figure legend (Line: 153-154).

4. Is *Tg2(NLS-Eos)* a typo? Even though it is listed as such on ZFIN, this name does not make sense. The original paper called this *fgf10b:nlsEOS*.

We named the transgenic line based on ZFIN, but we fully agree that *Tg2(NLS-Eos)* is confusing and not biologically informative. We have now updated the nomenclature throughout the manuscript to *Tg(fgf10b:nEOS)*, which is consistent with the original publication. In the Methods section, we added that this line was previously referred to as *Tg2(NLS-Eos)* so that the reader can link it to the ZFIN entry (Line: 391).

Reviewer 2 suggestions to the authors:

1. Lines 28-29. *Kimmel et al. (1995) provided the first details on gill development in zebrafish and should be added to this list of citations. (The paper is already cited in the methods section for its procedures.)*

Thank you for pointing this out. We have now added the citation of Kimmel et al. (1995) to the relevant section in the main text as suggested (Line: 26-27).

2. Lines 40-41. *The statement "and discouraging new investigators" is subjective and should be deleted.*

This statement has been removed.

3. Lines 67-68. *Did the authors actually observe blood flow? If not, I suggest rephrasing this statement.*

We did observe blood flow using the *Tg(gata1a:mCherry)* line. However, as this is an established feature of the adult filament vasculature and already mentioned in the introduction, we have removed the statement to avoid redundancy.

4. Line 69. *A word is missing. "Blood"?*

This sentence was removed along with the previous statement mentioned in point 3.

5. Figure 1H. *Upper left, should be "Gill raker". Bottom panel should be titled with the singular, "Lamella cross section". Same for Fig. 7E.*

Thank you for pointing this out. We changed it in both figures.

6. Line 83-84. *This statement is incorrect and must be revised: "uncovered a previously unrecognized, consistent medio-lateral asymmetry in filament length across all arches". This asymmetry is well-known to occur in the gills of nearly all fish species. Moreover, the medial vs. lateral asymmetry in developing gill filaments, and appearance of lamellae etc., has already been observed in another species: see Coughlan and Gloss (1984). Early morphological development of gills in smallmouth bass (*Micropterus dolomieu*). Canadian Journal of Zoology. 62: 951-958. I suggest rephrasing the statement with a focus on the evidence that the authors provide that shows the impacts of this asymmetry on gill development.*

We thank the reviewer for this valuable correction and for bringing the Coughlan and Gloss (1984) reference to our attention. We were indeed expecting this to be a previously described feature and looked for prior documentation in the literature, but were unfortunately unable to identify relevant sources. We have now removed the phrase “a previously unrecognized” from the sentence and cited the Coughlan and Gloss paper in the Discussion (Line: 298). That said, we believe our work provides new insights beyond what has been previously reported. While the medio-lateral difference has been described before, its developmental origin has not been described in detail, differences along the dorso-ventral axis were not explored, and the observed medio-lateral differences was not reported as lasting until adulthood. We have clarified this in the revised manuscript to better reflect both the existing literature and the novel aspects of our findings (Line: 297-300).

7. Line 101. Please label the tip cell in Fig. 3A (first panel). It is not clear.

The arrowhead in Fig. 3A (first panel) is intended to indicate the tip cell of the newly formed medial filament (Line: 98). We have clarified this in the figure legend by explicitly stating this in brackets.

8. Line 156. There is no arrowhead in Fig. 2A.

As noted in our response to Reviewer 1, point 3, this was an error on our part – the correct Figure should have been to Fig. 3A, not Fig. 2A. We have corrected this in the manuscript and clarified the description accordingly (Line: 153).

9. Lines 190-191. It is stated, “First signs of developing lamellae were observed at 4 dpf”, but the figure shows lamellae starting to develop in 5.5 dpf larvae. What were these first signs, and can an image at 4 dpf be provided, at least as a supplemental figure?

We appreciate the reviewer’s comment and want to clarify this point. The first signs of lamellae formation are indicated by the outgrowth of the outer marginal channel, as shown in Fig. 7A’ and Fig. 7B’ (lower panel). At 4 dpf, we observed identical looking endothelial tip cells only on medial filaments, which we explain in Fig. 8D, F. In contrast to 5.5 dpf, only a very limited number of lamellae were visible at 4 dpf. For this reason, we focused our analysis on the 5.5 dpf stage, where lamellae were more consistently present or in the process of forming. Nevertheless, we added a panel (A’) in Fig. S3, showing a representative image of 4dpf developmental stage of gill vasculature (Line: 189).

10. Lines 291-293. Regarding the statement: “While previous studies have provided detailed descriptions of the vascular anatomy of the developing zebrafish (Isogai et al., 2001), the description of the gill vasculature has remained sparse.” Can the authors qualify “sparse” and perhaps cite these studies? At least one study has reported some aspects of gill vasculature in zebrafish: Jonz and Nurse (2003). *J Comp Neurol.* 461:1-17.

We thank the reviewer for his comment and suggestion and agree that it is important to acknowledge earlier work addressing aspects of gill vasculature in zebrafish. In this context, we were referring specifically to the vascular atlas by Isogai et al. (2001), which, while comprehensive, includes only limited detail on the gill vasculature.

We also appreciate the reviewer highlighting the work of Jonz & Nurs, that offers valuable insights, particularly regarding the localization and physiology of neuroepithelial cells. However, we felt that other studies, such as Olsson et al. (2002), addresses the gill vasculature more directly. This study is also included in the highly comprehensive review by David H. Evans et al. (2005), which we have already cited in the introduction.

To address the reviewer’s suggestion, we have revised the paragraph to be more precise, clarifying that although previous studies have contributed valuable insights, the developmental dynamics of vasculature have remained largely underexplored in the existing literature (Line: 282-284).

Lines 391-392. The first part of this sentence is not accurate and must be revised: “With this study, *we establish the zebrafish gills as a new and valuable model to investigate complex biological processes*...”. Also, line 396 “this new model”. The zebrafish gill has been an important model in biology for decades. The authors will need to correct these statements.

We partially agree with the reviewer's comment. We fully acknowledge that the zebrafish gill has long served as an important model in biological research, as the reviewer rightly points out. Our intention was not to imply that the gill is *new* as a model per se, but rather to emphasize its potential as an *additional* and underutilized model for studying specific processes—such as angiogenesis, branching morphogenesis, and regeneration. We believe that this study will serve as both a foundation and a blueprint for such future research. To better reflect this, we now refer to the zebrafish gill as an additional model for studying these processes (Line: 378-383).

Line 459. Delete "several".

This has been modified accordingly.

Second decision letter

MS ID#: dev.204984R1

MS TITLE: A developmental atlas of zebrafish gills links early vascular patterning to adult architecture

AUTHORS: Virginie Lecaudey; Mathieu Preussner; Anna Mertens; Marion Basoglu

Dear Virginie,

I am happy with your revisions and pleased to tell you that your manuscript has been accepted for publication in Development, pending our standard publication integrity checks.